# Impact of Fecal Microbiota Transplant Formulations, Storage Conditions, and Duration on Bacterial Viability, Functionality, and Clinical Outcomes in Patients with Recurrent *Clostridioides difficile* Infection

**DOI:** 10.3390/microorganisms13030587

**Published:** 2025-03-04

**Authors:** Mohamed Shaheen, Chelsea McDougall, Leona Chan, Rose Franz, Karen Wong, Ryland T. Giebelhaus, Gwen Nguyen, Seo Lin Nam, A. Paulina de la Mata, Sam Yeo, James J. Harynuk, Sepideh Pakpour, Huiping Xu, Dina Kao

**Affiliations:** 1Division of Gastroenterology, Department of Medicine, University of Alberta, Edmonton, AB T6G 2X8, Canada; mshaheen@ualberta.ca (M.S.); cm35@ualberta.ca (C.M.); llchan1@ualberta.ca (L.C.); rose.franz@albertahealthservices.ca (R.F.); kwong3@ualberta.ca (K.W.); 2Department of Chemistry, University of Alberta, Edmonton, AB T6G 2G2, Canada; rgiebelh@ualberta.ca (R.T.G.); hanhnguy@ualberta.ca (G.N.); seolin@ualberta.ca (S.L.N.); delamata@ualberta.ca (A.P.d.l.M.); james.harynuk@ualberta.ca (J.J.H.); 3School of Engineering, University of British Columbia, Kelowna, BC V1V 1V7, Canada; sam.yeo@ubc.ca (S.Y.); sepideh.pakpour@ubc.ca (S.P.); 4Biostatstics & Health Data Sciences, School of Public Health, Indiana University, Indianapolis, IN 46202, USA; huipxu@iu.edu

**Keywords:** fecal microbiota transplantation, recurrent *Clostridioides difficile* infection, bacterial viability and functionality, FMT formulation and storage conditions, FMT efficacy

## Abstract

Fecal microbiota transplantation (FMT) is the most effective therapy for preventing recurrent *Clostridioides difficile* infection (rCDI). However, the impact of FMT formulations and storage conditions on bacterial viability, community structure, functionality, and clinical efficacy remains under-investigated. We studied the effect of different storage conditions on the bacterial viability (live/dead staining and cell sorting), community structure (16S rDNA analysis), and metabolic functionality (fermentation) of frozen and lyophilized FMT formulations. The clinical success rates of rCDI patients were correlated retrospectively with FMT formulations, storage durations, and host factors using the Edmonton FMT program database. Bacterial viability remained at 10–20% across various storage conditions and formulations and was comparable to that of fresh FMT. Live and dead bacterial fractions in both frozen and lyophilized FMT preparations exhibited distinct community structures. Storage durations, but not temperatures, negatively affected bacterial diversity. More short-chain fatty acids were found in the metabolomic profiling of in vitro fermentation products using lyophilized than frozen FMT. Clinical success rates in 537 rCDI patients receiving a single dose of FMT were not significantly different among the three formulations. However, longer storage durations and advanced recipient age negatively impacted clinical efficacy. Together, our findings suggest that FMT formulations and storage durations should be considered when establishing guidelines for product shelf life for optimal treatment outcomes.

## 1. Introduction

*Clostridioides difficile* infection (CDI) usually results from the dysbiosis of gut microbiota, particularly resulting from the use of antibiotics, creating an ecological niche that allows *C. difficile* to thrive and proliferate [1,2]. Recurrent *C. difficile* infection (rCDI) presents a challenge for clinicians because few therapeutic options exist. The traditional treatment of CDI includes antibiotics (e.g., metronidazole and vancomycin), which further exacerbate gut dysbiosis and lead to an increased risk of CDI recurrence. The fecal microbiota transplant (FMT) is currently the most effective treatment for CDI, irrespective of the route of administration [3], with clinical efficacies of >80% in randomized controlled trials (RCT) [4,5,6,7], and has been recommended by multiple practice guidelines for managing rCDI [8,9]. FMT formulations such as fresh, frozen, and lyophilized products are suggested to have similar clinical efficacy in preventing CDI recurrence in small studies [7,10,11,12,13]. However, questions remain regarding the superiority of one formulation over others and the appropriate “shelf life” for these products.

The mechanisms underlying FMT efficacy are not yet completely understood, but bacterial engraftment is thought to play a key role [14]. Thus, preserving bacterial viability, particularly anaerobes critical to gut health, is important for FMT manufacturing and storage processes [15,16,17]. Although there are guidelines for the manufacturing and storage of donor fecal material products, they are mostly based on expert opinions and not empirical research [18]. FMT manufacturing commonly involves the aerobic processing of donor fecal samples, which may significantly compromise the viability of anaerobic bacteria [19,20]. Furthermore, the freezing process, use of cryoprotectants, storage temperatures, and storage durations may differentially affect bacterial viability [21,22]. However, assessing bacterial viability is not straightforward. Traditional culture-based techniques are not always appropriate to evaluate bacterial viability because some bacteria are “unculturable” using existing protocols. Live/dead staining with fluorescence-activated cell sorting (FACS) is a convenient and rapid method to assess bacterial membrane integrity. However, FACS is costly and offers only partial insights, as it cannot evaluate the functional capacity of bacteria, which is crucial for therapeutic efficacy [23]. These considerations highlight the necessity of developing comprehensive methods that assess both bacterial viability and functionality, along with their combined impact on clinical outcomes.

To address these gaps, this study aims to (1) compare the bacterial viability and community structure of donor microbiota among FMT formulations (i.e., fresh, frozen, and lyophilized) stored at different temperatures and durations, (2) assess the metabolic functionality of these FMT products, and (3) correlate FMT formulations and storage durations with clinical outcomes.

## 2. Materials and Methods

### 2.1. Sample Information and Storage Conditions

Approximately 100 g of stool was collected from a registered stool donor in the Edmonton FMT program and processed within 4 h of collection. In brief, the stool sample was mixed with 200 mL water in a stomacher bag, homogenized, and divided into two 50 mL aliquots. The first 50 mL aliquot was mixed with trehalose (5% *v*/*v*, Swanson, Fargo, ND, USA) [24,25], and the second 50 mL was mixed with glycerol (10% *v*/*v*, Sigma G5516, Sigma Aldrich, St. Louis, MO, USA) [4]. The fecal slurry with glycerol was further divided into 1 mL aliquots, with 25 aliquots stored at −80 °C, and the remaining 25 stored at −20 °C. Similarly, the fecal slurry with trehalose was further divided into 1 mL aliquots, frozen overnight at −80 °C, and lyophilized for 48 h at −45 °C under ~300 mTorr. The lyophilized samples were divided equally into three Ziplock bags with desiccants and stored at −80, −20, and 4 °C, respectively. Samples were retrieved at 1 month, 3 months, 6 months, and 12 months for each experiment described below (Figure 1).

### 2.2. Sorting of Live and Dead Cells in Stored FMT Samples

#### 2.2.1. Sample Preparation for Live/Dead Staining

Prior to each experiment, lyophilized FMT (LFMT) samples were reconstituted in 1 mL of 0.9% saline and incubated at 37 °C for 30 min; frozen FMT (FFMT) samples were incubated at 37 °C for 30 min. The samples were then filtered through a 70 µm strainer and centrifuged at 10,000× *g* for 3 min. After removing the supernatant, the cell pellets were resuspended in 1 mL of 0.9% saline. For live and dead cell controls, the cell pellets were resuspended in 1 mL of PBS (1× phosphate-buffered saline) and 1 mL of 70% isopropyl alcohol, respectively, incubated at room temperature for 1 h, then centrifuged at 10,000× *g* for 3 min before resuspension.

#### 2.2.2. Live/Dead Staining of Bacteria

The bacterial cells in the FFMT and LFMT samples were stained using Live/Dead ^®^ BacLight^®^ Bacterial Viability and Counting Kit (L34856, Molecular Probes Inc., Eugene, OR, USA) following the manufacturer’s protocol, and live and dead cells were sorted with FACS. In brief, 977 µL aliquots of PBS were transferred into 1.5 mL microfuge tubes, to which 1.5 µL of SYTO9 stain (Component A), 1.5 µL of propidium iodide (Component B), and 10 µL of the sample was added; the samples were incubated at room temperature in the dark for 15 min. The microsphere standard (Component C) was resuspended by vortexing and sonication for 5–10 min before adding a 10 µL volume of microsphere suspension to each sample and mixing. The live cell control was stained with SYTO9, and the dead cell control was stained with propidium iodide.

#### 2.2.3. Live/Dead Bacterial Cell Sorting

FACS was performed on an LE-MA900 cell sorter (Sony Biotechnology, San Jose, CA, USA), and a 100,000-event number was used to set up the instrument and gating parameters with the live cell and dead cell control suspensions stained with SYTO9 and propidium iodide, respectively. Data was acquired using log scales for forward scatter and side scatter under green and red channels, and the protocol threshold was adjusted on the forward scatter parameter. Data acquisition and analysis were performed using LE-MA900FCP-Cell Sorter Software. FACS was carried out with a fixed sample volume of 1 mL per stained sample, and the number of collected events was not controlled to maximize the sorted cell recovery from individual samples. Fluorescent beads were used as an internal control and were distinguishable from stained bacteria. The percentages of live and dead bacteria were calculated using the sort-count data.

### 2.3. 16S rDNA Analysis

#### 2.3.1. DNA Extraction and 16S rDNA Amplicon Library Preparation

DNA was extracted from the live and dead cell fractions using the PowerFecal Pro^®^ DNA Isolation Kit (Qiagen, Venlo, The Netherlands); 16S rDNA amplicon libraries were prepared targeting the V3–V4 variable region as described by Holm and colleagues (Appendix A) [26]. The quality of the amplification was evaluated with Invitrogen E-gel electrophoresis. Libraries were normalized and pooled with the SequalPrep™ Normalization Plate Kit (Applied Biosystems™, Westminster, CO, USA). Library fragments of ~620 bp were selected with SPRI beads, and the pooled library was sequenced on an Illumina MiSeq 600-cycle cartridge (MiSeq Reagent Kit v3, San Diego, CA, USA).

#### 2.3.2. Bioinformatics Analysis of 16S rDNA Amplicon Sequences

De-multiplexed raw Illumina MiSeq sequence data were initially assessed for quality using FastQC (version 0.11.9). The primers were removed from the reads using the cutadapt function [27], and quality trimming and filtering of the reads was performed using DADA2 software (version 1.26) [28]. The PhiX reads were removed from both forward and reverse reads, followed by filtering based on parameter maxEE = c(2,3). The good-quality reads were merged, analyzed, and assigned to taxonomic classification against the Mothur-formatted SILVA database (Release 138.1) using Mothur (version 1.48.1) [29]. Because the samples showed large variations in sequencing depth, rarefaction using minimal reads (4737) per sample together with total sum scaling was performed on Microbiomeanalyst2 [30] to compare the bacterial diversities of the samples. The Shannon index was used to determine differences in α-diversity with a non-parametric Hutcheson *t*-test. The Bray–Curtis dissimilarity metric was used to determine the differences in community structures between the FMT sample groups (β-diversity) with a 2D principal coordinates analysis (PCoA) plot visualization and to perform the Ward linkage-based clustering. The statistical significance of the differences in the β-diversity of the FMT sample groups was evaluated using permutation multivariate analysis of variance (PERMANOVA) with 9999 permutations. Linear discriminant analysis–effect size (LEfSe) was carried out using the Kruskal–Wallis test with a correlation threshold of 0.3 and an adjusted *p*-value cutoff at 0.05, and clustering of the FMT samples was performed using Ward’s linkage of the Bray–Curtis distances at the genus level. Univariate (DESeq2) and multivariable regression analyses (MaAsLin2) were used to evaluate the association between the bacterial community and storage temperature. All statistical analyses were performed using Mothur or Microbiomanalyst2, and the data visualizations were performed in R (v4.4.1) using the tidyverse packages [31].

### 2.4. In Vitro Fermentation of Fibers by Stored FFMT and LFMT

In vitro fermentation of fibers was carried out in 2.5% brain heart infusion (BHI) broth supplemented with a fiber nutrient (arabinoxylan or inulin). A 4% (*v*/*v*) suspension of each FMT sample was prepared in sterile PBS, and 2.5 mL of the suspension was added to each fermentation tube containing 2.5 mL of 5% BHI supplemented with 0.25% arabinoxylan or inulin. Fermentation was performed in triplicate for each FMT sample, with PBS as the fiber-free blank control. The fermentation tubes were incubated anaerobically for 48 h at 37 °C on a shaker at 125 rpm. After fermentation, 1 mL aliquots were taken from each replicate and centrifuged at 20,000× *g* for 20 min. The fermentation supernatants (400 µL) were withdrawn from each of the three replicates, combined to make a composite sample, mixed with 300 µL of 25% phosphoric acid, and stored at −80 °C until further analysis.

### 2.5. Metabolomic Profiling of In Vitro Fermentation Products

#### 2.5.1. SPME-GC×GC-TOFMS Untargeted Metabolomics

The metabolomic profiles, with an emphasis on short-chain fatty acids (SCFAs), of the composite fermentation products were analyzed by sampling the headspace with solid-phase microextraction (SPME) followed by comprehensive two-dimensional gas chromatography–time-of-flight mass spectrometry (GC×GC-TOFMS) based on untargeted metabolomics (Appendix A).

The chromatographic data was processed using LECO ChromaTOF^®^ BT software (v5.55.41). Retention indices were computed based on the elution times of the linear alkanes. All chromatographic peaks were searched against the NISTMS 2020 Libraries, with a minimum mass spectral similarity of 700 to assign a putative ID. All annotated metabolites were putatively identified to the Metabolomics Standards Initiative level 2, unless otherwise mentioned [32]. Mass spectra and retention indices for linear SCFAs of interest (acetic acid to decanoic acid) were tabulated (Appendix A). In lieu of reference standards, putative identities were assigned to linear SCFAs using mass spectral library and retention index matching. A pooled quality control (QC) sample was included with each batch, and quality checks were manually performed by inspecting QC samples, replicate injections, and blanks. Following quality checks, all sample chromatograms were aligned into one cohesive peak table with annotation using ChromaTOF^®^ Sync 2D (v2.0.9.1-beta; LECO) with a S/N of 1000.

#### 2.5.2. Data Analysis and Chemometrics

The aligned peak table (*n* = 32) was imported into MATLAB^®^ R2022a (The MathWorks Inc., Natick, MA, USA) for statistical analysis. First, the peak table was normalized to the total useful peak area (TUPA) [33]. TUPA is a well-established, data-driven normalization strategy which has been implemented in numerous GC×GC-TOFMS studies. Peak tables were labeled based on fiber, temperature, time, and formulation, and then split into separate peak tables based on the experimental design. Principal component analysis (PCA) models were generated on the auto-scaled peak tables before and after FS using PLS_Toolbox 9.0 (Eigenvector Research, Manson, WA, USA) in the MATLAB^®^ environment. Outliers with high Hotelling T^2^ or Q residuals were removed. PERMANOVA was performed on the auto-scaled peak tables for each comparison in R (v4.4.1) with the vegan package (v2.6-4), using Euclidean distance and 99,999 permutations and a significance threshold of *p* < 0.05 [34].

### 2.6. FMT Formulations, Storage Conditions, and Clinical Outcomes in rCDI Patients

To assess the clinical efficacy associated with different FMT formulations and storage durations, a retrospective analysis was conducted on the clinical outcomes of FMT recipients in the Edmonton FMT program between 2013 and 2022 (REB approval Pro00101823), where metadata were available for patient age, sex, and number of CDI episodes, FMT formulations and storage durations, and treatment outcomes. Eligibility for receiving FMT was considered to be the following: (1) at least two CDI recurrences (i.e., a total of three CDI episodes) or (2) at least one CDI recurrence requiring hospitalization. Fresh FMT was used from October 2012 until February 2017, FFMT was used from January 2013 until September 2022, and LFMT was subsequently introduced to the FMT program in March 2018. Each FMT dose was manufactured with at least 25 g of donor stool. Frozen and lyophilized FMT products were stored at −80 °C.

Following FMT, patients were followed for at least 8 weeks. Treatment success was defined as no recurrence of CDI 8 weeks following FMT. The clinical outcome data were analyzed in R (v4.4.1), and non-parametric regression was performed to estimate the association of different formulations and storage durations with clinical outcomes. Pearson’s chi-square test was used for categorical variables, and the Wilcoxon rank sum test was used for continuous variables. A logistic regression model was used to determine and compare the association of factors (e.g., FMT formulations and storage conditions; patient age, sex, antibiotic use prior to CDI, and number of CDI episodes) with treatment outcomes.

## 3. Results

### 3.1. Viability of Bacterial Populations Stored at Different Temperatures and Durations

After aerobic processing, approximately 20% of the bacterial population in the donor fecal sample remained viable, which was used as the baseline value to compare the effects of formulation (frozen or lyophilized), storage duration (3, 6, 9, or 12 months), and storage temperature (frozen [−20 °C or −80 °C] or lyophilized [4 °C, −20 °C, or −80 °C]) on bacterial viability. The viable proportions of cells did not decrease over the 12-month study period when frozen and stored at −20 °C or −80 °C when compared with the viability of fresh FMT. In contrast, the viable proportions of bacteria decreased by 13% immediately after lyophilization. After this initial decrease, populations remained fairly stable after 1 month of storage at −80 °C (10%), −20 °C (15%), or 4 °C (13%) and did not diminish further within the 12-month study period (Figure 2).

### 3.2. Taxonomic Analysis of the Bacterial Communities of Stored FMT Samples

DNA isolated from 34 dead and live cell fractions from the 17 FMT samples generated 1,107,543 reads from Illumina MiSeq sequencing after removing the primers and performing the quality trimming and filtering. Out of 34 cell fraction samples, the dead fraction of the LFMT sample stored at −20 °C for 12 months produced only one read and, therefore, was excluded from further analysis.

#### 3.2.1. Bacterial Communities of Live and Dead Cell Fractions

The bacterial compositions in the live and dead cell fractions, with a minimum relative abundance of 0.2% across phylum to genus levels, exhibited distinct variations in their distributions (Figure 3).

The community-level comparison of bacterial populations found statistically significant differences, indicated by Shannon diversity (α-diversity) indices, between the live and dead cell fractions (*p* < 0.05) (Figure 3a). The β-diversity index analysis with PCoA plotting of Bray–Curtis distances also demonstrated statistically significant differences between the bacterial communities of these two cell fractions (*p* < 0.05) (Figure 3b). Moreover, the clustering analysis portrayed a clear separation between live and dead cell fraction clusters (Appendix A). The dendrogram also showed that most of the LFMT and FFMT samples formed their own respective clusters, suggesting that the microbial diversity within the LFMT and FFMT samples had distinct community features. Firmicutes and Actinobacteriota were the two most abundant phyla in both cell fractions. However, Actinobacteriota exhibited a higher abundance in the live cell fractions than in the dead cell fractions, while the opposite trend was observed for Firmicutes. At the genus level, *Blautia*, *Bifidobacterium*, *Dorea*, and *Faecalibacterium* were more abundant in the live cell fractions, whereas *Fusicatenibacter*, *Anaerostipes*, and members of *Lachnospiraceae* were more abundant in the dead cell fractions (Figure 3c–g). LEfSe analysis identified all the statistically significant differentially abundant genera in the live versus dead cell fractions (Appendix A).

#### 3.2.2. Variations in Live Bacterial Communities Due to Formulations and Storage Conditions

The live cell fractions of the FMT samples were further analyzed to understand the impact of formulation and storage duration on the bacterial community structures. Although statistically significant differences in α-diversity (Shannon) and β-diversity (Bray–Curtis distances) indices were observed (*p* < 0.05) in the live bacterial communities between the FFMT and LFMT samples, the large differences in Shannon diversity at baseline (M0) and at 3 months (M3) between FFMT and LFMT could have skewed the overall comparison because only single data points were available (Figure 4a–c).

Shannon diversity indices consistently decreased over time and were the highest at M0 and lowest at 12 months (M12), suggesting that prolonged storage duration negatively affected bacterial diversity. There were no statistically significant differences in bacterial communities resulting from storage temperatures identified by univariate analysis (DESeq2), except for *Deinococcus*, which exhibited significantly higher abundance in the samples stored at −80 °C than at the other storage temperatures. However, this finding may be influenced by the exceptionally high abundance of this genus in a single sample (FFMT sample stored at −80 °C for 3 months), which may have skewed the results. The multivariable regression analysis (MaAsLin2), when adjusted for storage duration as covariate, did not find statistically significant differences (*p* < 0.05) in bacterial communities resulting from storage temperature (Appendix A). Although the PCoA plot of the β-diversity indices suggested small differences in the live bacterial communities between the FFMT and LFMT samples, PERMANOVA analysis confirmed a statistically significant difference between the sample groups (*p* < 0.05) (Figure 4a–c). When the relative abundance data were plotted, minor differences were observed at the phylum level between the FFMT and LFMT samples. However, at the genus level, *Blautia* was more abundant in FFMT, whereas *Bifidobacterium*, *Fusicatenibacter*, *Anaerostipes*, *Dorea*, *Faecalibacterium*, and *Romboutsia* were found in higher relative abundance in LFMT samples (Figure 4d,e). The relative abundance of Firmicutes consistently but not significantly decreased over the 12-month study period for both the frozen and lyophilized formulations. Only minor variations were observed in the relative abundances of the most dominant genera over the storage durations (Figure 4f,g). Notably, the proportion of low-abundance genera decreased and the relative abundance of *Bifidobacterium*, *Fusicatenibacter*, *Anaerostipes*, and *Dorea* increased with storage duration.

### 3.3. Metabolic Profiles of Anaerobic In Vitro Fiber Fermentation Products

A total of 361 metabolites were identified in the fermentation supernatants. One outlier sample was identified and removed on the basis of its high Hotelling T^2^ value (Appendix A). Visual inspection and comparison of media and reagent blanks to pooled QC samples confirmed that the detected metabolites were only present in the samples (Appendix A). Peaks in blanks were caused by siloxanes from the SPME fiber and the internal standard (Appendix A). Blanks had <150 identifiable peaks, whereas the pooled QC samples had >300 identifiable peaks. Normalization by TUPA corrected for sample-related variations and ensured that the compounds from the SPME fiber did not influence the statistical analysis [33]. Batch labels were applied to PCA score plots generated using all 361 variables and just the putatively identified linear SCFAs, both normalized to TUPA (Appendix A). The separation of samples in batch 1 and 2 in Appendix A is the result of those batches containing samples that projected away from other samples based on their chemical characteristics rather than batch effects alone (Appendix A). This is further reinforced by Appendix A, where there is no batch effect when considering linear SCFAs.

Separation along PC1 was observed between FFMT and LFMT when considering all variables (Appendix A). The comparisons of time and formulations were shown to be statistically significant by PERMANOVA (*p* < 0.05) when considering all variables (Appendix A). When the linear SCFAs (acetic acid to decanoic acid) were considered, only the difference between frozen and lyophilized formulations was statistically significant (PERMANOVA, *p* < 0.05), with the lyophilized formulations producing more SCFAs. All SCFAs had mass spectra (Appendix A), and retention indices closely matching library values. PCA score plots generated using only SCFAs showed clear separation along PC1 between FFMT and LFMT formulations (Figure 5d), and PC2 described changes in SCFA levels resulting from storage duration (Figure 5c).

PC1 also captured the variance describing temperature-based differences, with 4 °C projecting to the right and the frozen FMTs (−20 and −80 °C) projecting to the left (Figure 3b). The biplot (Figure 5e) showed that all linear SCFAs, except propanoic acid, were correlated with LFMT.

### 3.4. Correlation of FMT Formulation and Storage Duration with Clinical Outcomes

The cohort consisted of 537 patients with rCDI who received either fresh, frozen, or lyophilized FMT. The baseline characteristics of these FMT recipients are summarized in Table 1.

Overall, no statistically significant differences in success rates were observed between the patients who received fresh, frozen, or lyophilized FMT. As shown in Table 1 and Table 2, both adjusted odds ratios and *p*-values of the success rates of FFMT and LFMT were not statistically different from those of fresh FMT.

The only two statistically significant factors contributing to treatment outcomes were patient age and FMT storage duration. Age was negatively correlated with successful outcomes, independent of FMT formulation (Table 1 and Figure 6).

Specifically, success rates decreased with increasing age but plateaued at 75 years. FMT storage duration negatively impacted clinical success rates, with longer storage durations associated with lower treatment success; the rate of decline was more pronounced for the lyophilized formulation than for the frozen formulation (Table 2 and Figure 6).

## 4. Discussion

FMT is a guideline-recommended therapy to prevent rCDI and a promising investigational treatment in dysbiosis-associated conditions such as ulcerative colitis [3,35,36]. Therapeutic efficacy is thought to be mediated by bacterial engraftment and bacterially derived metabolites [14]. We found that while storage duration and temperature did not affect bacterial viability, a gradual decrease in bacterial diversity was observed over the 12-month study period in both frozen and lyophilized formulations. Although similar levels of highly abundant bacteria were observed, differences in community structure existed between these formulations. The untargeted metabolomics analysis of fiber fermentation products also found significant differences between formulations, including SCFA. These compositional changes over time and functional differences between formulations aligned with the observed decreasing clinical success rate with longer storage duration, especially for lyophilized FMT, in our retrospective rCDI patient cohort. Despite this, treatment with both FFMT and LFMT still achieved over 75% success when stored for up to 250 and 140 days, respectively.

Most FMT manufacturing protocols use aerobic processing, including our own, which likely accounted for the low proportion of viable cells (20%) at the start of our experiments. Papanicolas and colleagues also found that approximately 20% of the bacteria remained viable after the aerobic processing of FMT via 16S rDNA qPCR in conjugation with propidium monoazide treatment. They also found that anaerobic processing improved the viability of obligate anaerobes and increased the proportion of total viable cells to approximately 50%; however, a single freeze–thaw cycle at −80 °C for 48 h reduced the viability to 23%, despite using 10% glycerol as a cryoprotectant [22]. In contrast, using culturomics, Fouhy and colleagues found no differences in anaerobic bacterial counts between fecal samples that were fresh, snap frozen, or stored at −80 °C for up to 7 days, even without using a cryoprotectant [37]. Although the use of cryoprotectants is common to minimize the effects of freezing on bacterial membrane integrity, no differences in bacterial functionality (e.g., SCFA production) were reported for fecal samples stored at −80 °C for 106 days with or without cryoprotectants such as 5% DMSO (dimethyl sulfoxide) or DMSO–media mix (DMSO + tryptic soy broth with trehalose) [38]. The vastly different results from various studies highlight the challenges with assessing bacterial viability. It is not known if a higher viable bacterial population resulting from anaerobic processing or the addition of a cryoprotectant to preserve the bacterial membrane integrity would have a significant impact on clinical efficacy, because aerobically processed FMT already has very high cure rates (>80%) for rCDI [3,36,39]. However, these processing strategies could potentially be relevant when FMT is used for other indications beyond rCDI.

To further characterize which bacteria survive FMT manufacturing and storage conditions, we used 16S rDNA analysis to reveal fundamental differences in the bacterial populations between the live and dead cell fractions of the FMT samples. Consistent with our study, Bellali and colleagues also found a higher abundance of Actinobacteriota and a lower abundance of Firmicutes in the live cell fractions than in the dead cell fractions of FMT samples using live/dead staining with FACS [40], although Firmicutes was the most abundant phylum in both fractions. Similar trends in the relative abundance of these phyla, as determined by 16S rDNA of live bacteria (i.e., propidium iodide-treated samples), were reported for human fecal samples stored for up to 1 year [41]. In the live cell fractions, we identified high relative abundances of *Blautia* and *Fusicatenibacter* (*Firmicutes phylum*) and *Bifidobacteria* (*Actinobacteriota phylum*); these are predominant genera in gut microbiota and are recognized for their roles in maintaining gut mucosal functions and SCFA production [42,43,44]. Additionally, genera such as *Agathobacter* and *Dorea*, which are also involved in SCFA production, were present at significantly higher abundances in the live cell fractions than in the dead cell fractions. We further demonstrated that, although the relative abundances of dominant genera remained stable, the relative abundance of SCFA producers like *Faecalibacterium*, *Agathobacter*, and many other low-abundance genera decreased in the live cell fractions over 12 months. This may not be entirely surprising, because non-spore formers may not be as resilient as spore-forming Firmicutes to FMT manufacturing processes and storage conditions. These microbial community changes may contribute to the decreasing clinical efficacy of FMT products after longer storage duration.

The recommended storage durations for FMT products vary, and these recommendations are not necessarily based on evidence. For example, the International Consensus Conference on stool banking had previously recommended storing fecal materials at −80 °C for up to 2 years [18]. More recently, the British Society of Gastroenterology advised limiting the storage duration to no longer than 1 year at −70 °C [45]. A quality assurance study from the United States-based, non-profit public stool bank OpenBiome analyzed data from 257 facilities and 1924 frozen FMTs; they reported a cure rate of 83.8% using products stored for a mean of 139 days, with no statistically significant impacts of storage duration on clinical outcomes [46]. In contrast, our results showed diminishing clinical efficacy with longer storage duration, and the rate of decline was much more pronounced with lyophilized FMT compared with frozen FMT. As such, the optimal “shelf life” of FMT products may be formulation- and indication-dependent and is likely between 6 and 12 months. It should be noted that our retrospective study used FMT from multiple donors in clinical care. While we did not formally examine whether clinical efficacy is donor dependent, OpenBiome, a public stool bank, reported no statistically significant difference in efficacy amongst 51 donors whose stools were used to treat 2050 patients [47].

Although the ideal FMT formulation has not been determined, results from this and other studies suggest that aerobically processed FMT that is lyophilized and stored at −80 °C or 4 °C would likely have similar efficacy as frozen products and possess similar biologically important functions for preventing CDI recurrence. Indeed, a meta-analysis examining four RCTs and four observational studies reported the efficacy of fresh, frozen, and lyophilized FMT to be 95%, 88%, and 83%, respectively, with no statistically significant differences among groups [48]; these results are congruent with our findings. Lyophilized products are more practical, as they can be stored at 4 °C or room temperature and therefore require less infrastructure support than frozen FMT.

In addition to the characteristics of FMT, we also found that advanced age negatively impacted clinical outcomes. Rajita and a colleague recently reported that increasing age was associated with a higher CDI recurrence rate following treatment with a defined microbial consortium VE303 [49]. In the same study, they further identified lower bacterial engraftment in these older recipients of VE303. Since the diversity and functionality of the gut microbiota may decline with age [50], further consideration may need to be given to this group of patients, such as using multiple FMT doses or choosing a product with shorter storage durations.

Our study was limited by the lack of replicates for all the experiments, except for the in vitro fermentation experiments. Additionally, only a single stool donation from a single donor was used in this study, and metabolomics analysis was performed only on volatile compounds that could have been affected by the lyophilization process or the preanalytical stage of the metabolomic analysis. Furthermore, the FMT samples used in the in vitro experiments were not used to treat patients, which limited our ability to directly assess clinical efficacy. Therefore, future studies that use the same FMT samples for in vitro assessment described in this study and for patient care would better correlate outcomes. At the same time, using multiple donor samples for in vitro assessment would allow the generalizability of our findings. Our clinical efficacy data were derived from retrospective data, with relatively few data points for the lyophilized FMT cohort with storage durations beyond 150 days, limiting our ability to accurately assess an ideal “shelf life” for lyophilized FMT. We did not assess how other key microbial functions (e.g., bile acid conversion and other antibacterial peptide production) are affected by FMT manufacturing, formulation, or storage conditions, or how these functions impact clinical efficacy. Future research should address these shortcomings; preserving bacterial viability and functionality in FMT is not only relevant in the management of rCDI but will likely be even more important in other dysbiosis-associated indications.

In conclusion, this study enhances our understanding of how different FMT formulations and storage conditions could impact bacterial viability, community compositions, and functionality. While previous studies and this study suggested comparable clinical efficacy across fresh, frozen, and lyophilized formulations, our results highlight how these factors influence bacterial viability, community structures, and functionality and how the physical and biological factors may potentially affect clinical outcomes. The implication of this finding is that the trade-off for the convenience of lyophilized FMT comes with reduced shelf life, and that FMT inventory would need to be closely monitored.

## Figures and Tables

**Figure 1 microorganisms-13-00587-f001:**
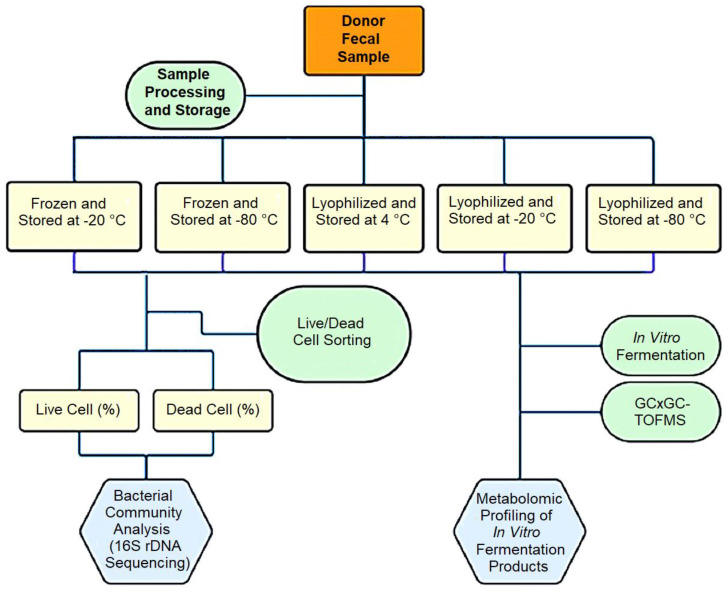
Schematic diagram of the study design: A single fecal donation was processed aerobically and preserved using two methods: freezing and lyophilization. The samples were then stored under various temperatures for up to one year and evaluated at three-month intervals. Analyses included bacterial viability using live/dead staining and fluorescence-activated cell sorting (FACS), bacterial community structure by 16S rRNA gene amplicon sequence analysis, and microbial functionality through in vitro fermentation of fiber products, followed by a GC×GC-TOFMS analysis of volatile metabolites, with a particular focus on short-chain fatty acids.

**Figure 2 microorganisms-13-00587-f002:**
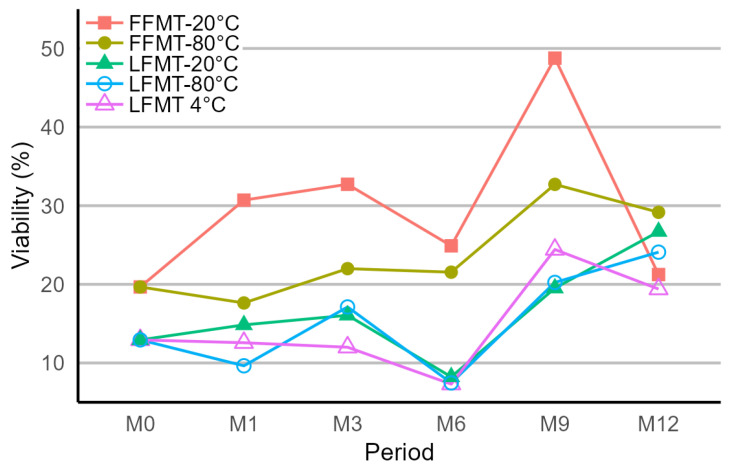
Proportion of viable bacteria in frozen fecal microbiota transplantation (FFMT) and lyophilized FMT (LFMT) samples stored at different temperatures and storage durations (M = month) determined by live/dead straining coupled with fluorescence-activated cell sorting.

**Figure 3 microorganisms-13-00587-f003:**
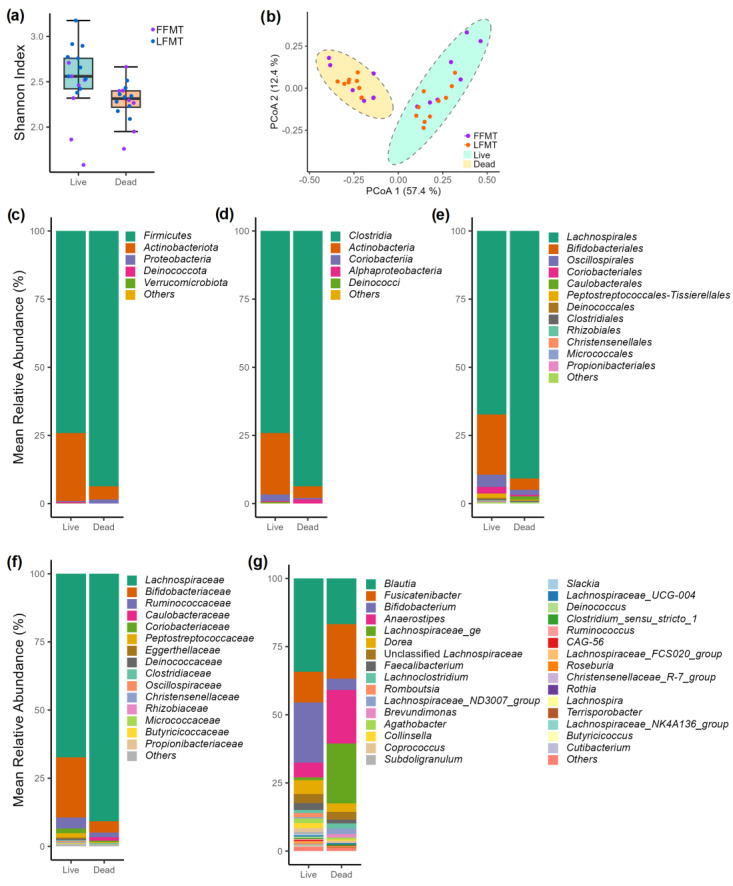
Bacterial community structures and diversity indices in live and dead cell fractions of fecal microbiota transplantation (FMT) samples stored under different conditions. (**a**) The α-diversity (Shannon) indices of live and dead fractions of the FMT samples (frozen [FFMT] + lyophilized [LFMT]) were statistically significantly different between groups (*p*-value < 0.05). (**b**) The principal coordinate analysis (PCoA) plot of the β-diversity indices (Bray–Curtis dissimilarity indices) of live and dead cell fractions of the FMT samples exhibited significant differences in the bacterial communities between these groups (*p*-value < 0.05). The FFMT and LFMT samples are shown in different colors in the α-diversity and β-diversity plots. Mean relative abundance of bacteria in all FMT samples (FFMT + LFMT) at (**c**) phylum level, (**d**) class level, (**e**) order level, (**f**) family level, and (**g**) genus level.

**Figure 4 microorganisms-13-00587-f004:**
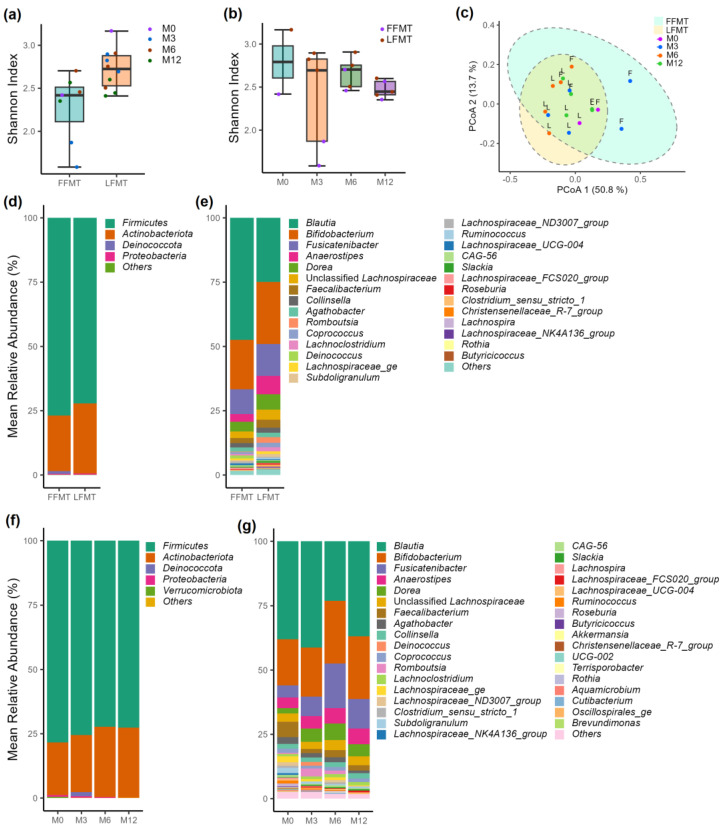
Bacterial community structures and diversity indices in live cell fractions of fecal microbiota transplantation (FMT) samples stored under different conditions. (**a**) Shannon indices of live cell fractions in frozen (FFMT) + lyophilized (LFMT) samples were significantly different between groups (*p*-value < 0.05). (**b**) Shannon indices of live cell fractions at different time points were not significantly different between the groups (*p*-value > 0.05). (**c**) The principal coordinate analysis plot of the β-diversity indices (Bray–Curtis dissimilarity indices) of the live cell fractions of the FFMT (F) and LFMT (L) samples had significantly different bacterial communities (*p*-value < 0.05). Mean relative abundance of bacteria in frozen FFMT and LFMT at the (**d**) phylum level and (**e**) genus level. Mean relative abundance of bacteria at different time points at the (**f**) phylum level and (**g**) genus level.

**Figure 5 microorganisms-13-00587-f005:**
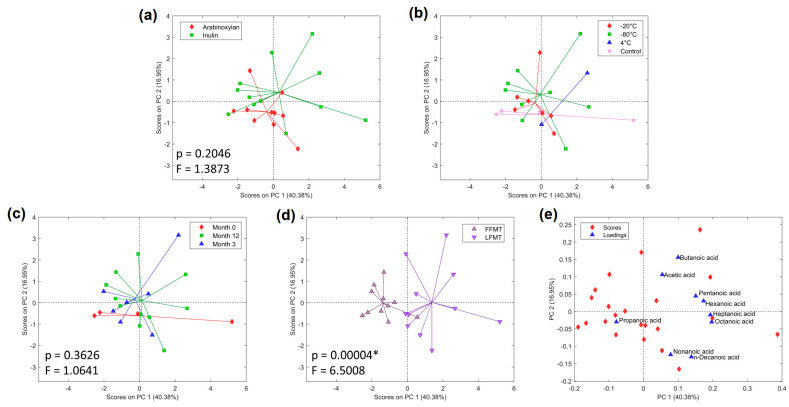
Principal component analysis (PCA) scores generated using just short-chain fatty acids (SCFAs): (**a**) arabinoxylan vs. inulin; (**b**) temperature; (**c**) time (month 0, 3, and 12); (**d**) frozen fecal microbiota transplantation material (FFMT) vs. lyophilized FMT material (LFMT); (**e**) biplot showing scores (red diamonds) and loadings (blue triangles). Scores represent individual samples, and loadings represent individual variables and how they contribute to the principal components. * denotes statistical significance (*p* < 0.05). Summary statistics cannot be computed for (**b**) due to low statistical power.

**Figure 6 microorganisms-13-00587-f006:**
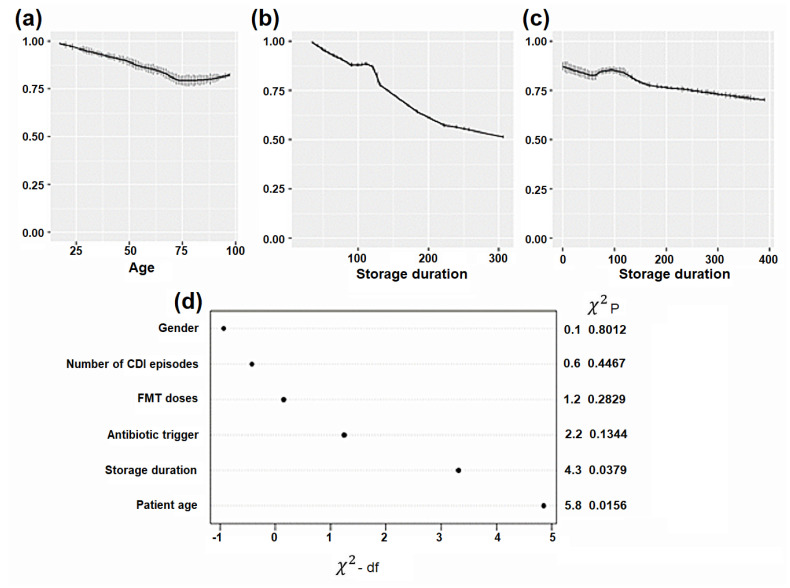
Clinical success rate of fecal microbiota transplantation (FMT). (**a**) The age of patients negatively affected FMT efficacy. The efficacy of frozen FMT (FFMT) and lyophilized FMT (LFMT) materials gradually decreased with increasing storage durations. Greater than 75% success rates were observed with (**b**) LFMT stored for up to 140 days and (**c**) FFMT stored for up to 250 days. (**d**) The multivariate analysis also showed that the age of the patient and FMT storage duration had the greatest negative impacts on clinical success rates.

**Table 1 microorganisms-13-00587-t001:** Fecal microbiota transplantation (FMT) recipient characteristics and clinical success rates.

		Fresh FMT (*n* = 33)	Frozen FMT (*n* = 406)	Lyophilized FMT (*n* = 98)	*p*-Value
Age in years, mean (SD)		61.2 (20.5)	66.5 (17.5)	62.2 (17.5)	0.036 *
Sex	Female	18 (54.5%)	241 (59.4%)	62 (63.3%)	-
	Male	15 (45.5%)	165 (40.6%)	36 (36.7%)	0.64
Number of CDI episodes prior to FMT, median (IQR)		3 (3–3)	3 (3–4)	3 (3–4)	0.14
Antibiotic trigger prior to CDI		32 (97.0%)	356 (87.7%)	89 (90.8%)	0.21
Storage duration in days, median (IQR)		-	46.5 (21–107)	100 (69–127)	<0.001 *
Success rate, mean %		83.0	90.9	85.7	0.36
Success rate, above 75%		NA	250 days	140 days	-

*p*-Values were calculated based on Pearson’s chi-square test for categorical variables and the Wilcoxon rank sum test for continuous variables. * denotes statistical significance (*p* < 0.05). SD = standard deviation, IQR = interquartile range, CDI = *Clostridioides difficile* infection and NA = not applicable.

**Table 2 microorganisms-13-00587-t002:** Adjusted odds ratios for the success of lyophilized (LFMT) and frozen (FFMT) fecal microbiota transplantation material using a multivariable logistic regression model.

		Odds Ratio (95% CI)		*p*-Value	
		FFMT	LFMT	FFMT	LFMT
Age in years		0.979 (0.962, 0.996)	1.004 (0.972, 1.038)	0.017	0.8
Sex	Female	1.135 (0.667, 1.931)	2.230 (0.688, 7.231)	0.64	0.18
	Male	Reference	Reference	-	-
Number of CDI episodes prior to FMT		1.123 (0.891, 1.415)	1.141 (0.590, 2.206)	0.33	0.7
Antibiotic trigger prior to CDI		0.448 (0.143, 1.400)	2.508 (0.418, 15.052)	0.17	0.31
Storage duration		0.997 (0.994, 0.9998)	0.990 (0.981, 0.999)	0.033 *	0.038 *

* denotes statistical significance (*p* < 0.05). CDI = *Clostridioides difficile* infection, CI = confidence interval.

## Data Availability

The raw sequencing data have been deposited into the Sequence Read Archive of NCBI [http://www.ncbi.nlm.nih.gov/sra], and the accession number is PRJNA1228141.

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
