# Peer review of "Impact of Fecal Microbiota Transplant Formulations, Storage Conditions, and Duration on Bacterial Viability, Functionality, and Clinical Outcomes in Patients with Recurrent *Clostridioides difficile* Infection"

_microorganisms, 2025, doi:10.3390/microorganisms13030587_

Round 1
Reviewer 1 Report
Comments and Suggestions for Authors
It is an interesting and well presented ms. However, the authors could clarify why they chose this specific normalization method over others and how they addressed batch effects, including whether QC samples were used. It would also be helpful to specify if SCFA identifications were confirmed with reference standards and retention indices. Lastly, considering the potential concentration effect of lyophilization on SCFA levels, the authors could note if any corrections were applied.
Reviewer 2 Report
Comments and Suggestions for Authors
The manuscript entitled "Impact of Fecal Microbiota Transplant Formulations, Storage Conditions, and Durations on Bacterial Viability, Functionality, and Clinical Outcomes in Patients with Recurrent Clostridioides difficile Infection”, studied the effect of different storage conditions on bacterial viability (live/dead staining and cell sorting), community structure (16S rDNA analysis), and metabolic functionality (fermentation) of frozen and lyophilized FMT formulations. Despite the good structure of the manuscript, there are some concerns which must be addressed.
- Line 25; please write in vitro in italic.
- Please add more references to materials and methods section
- Please design your tables following previous publications at the journal.
- It will be better to add “in conclusion” paragraph at the end of the discussion section.
- Given that your study showed diminishing clinical efficacy with longer storage duration, can you clarify how the results align with previous findings from OpenBiome, which reported no significant impact of storage duration on clinical outcomes?
- You observed a gradual decrease in bacterial diversity over the 12-month study period. How might this reduction in diversity specifically contribute to the decreasing clinical success rates, and what thresholds of diversity do you consider critical for efficacy?
- Given that only a single stool donation from one donor was used, how might this limit the generalizability of your findings? Do you plan to investigate the variability in FMT efficacy using multiple donors?
- Considering the potential for lyophilized FMT to be stored at higher temperatures, what logistical considerations should clinics take into account when transitioning from frozen to lyophilized formulations?
